# Supportive Implant Therapy (SIT): A Prospective 10-Year Study of Patient Compliance Rates and Impacting Factors

**DOI:** 10.3390/jcm9061988

**Published:** 2020-06-25

**Authors:** Julia Mitschke, Stefanie Anna Peikert, Kirstin Vach, Eberhard Frisch

**Affiliations:** 1Northern Hessia Implant Center, D-34369 Hofgeismar, Germany; julia.mitschke@t-online.de; 2Department of Operative Dentistry and Periodontology, Faculty of Medicine, Medical Center, University of Freiburg, D-79106 Freiburg, Germany; stefanie.peikert@uniklinik-freiburg.de; 3Institute for Medical Biometry and Statistics, Faculty of Medicine and Medical Center, University of Freiburg, 79104 Freiburg, Germany; kv@imbi.uni-freiburg.de

**Keywords:** dental implants, compliance, aftercare, long-term results

## Abstract

The main objective of this study is to present patient compliance rates and influential factors for regular attendance in a systematic implant aftercare program (Supportive Implant Therapy; SIT) within a 10-year observation period. From 2005 to 2008, we identified 233 patients with 524 implants and implant-supported restorations at the study center. They had been instructed to attend an SIT program with 3-month recall intervals. A 2019 clinical prospective cohort study on 10-year compliance rates was performed. Data were assessed yearly in regression analyses to identify influential factors. Noncompliance rates increased during the period (4.8%, year 1; 39.7%, year 10). Total noncompliance was observed in four patients (1.7%) with 10 implants. “Age,” “Gender,” “Diabetes”, and “Surgical case complexity” showed no correlation with patient compliance. “Smoking” and “Cardiovascular diseases” significantly influenced patients in one of ten years, while “Number of implants per patient”, “Type of implant-supported prostheses”, and “Pre-existing experience in a prophylaxis program” reached significance after several years. When patients with implant-supported restorations are strongly recommended and frequently remotivated to comply with an SIT program with 3-month recall, an approximately 60% compliance rate after 10 years is achievable. Previous prophylaxis program experience, increased number of implants per patient, and removable implant-supported prostheses may be strong influential factors for increased patient compliance.

## 1. Introduction

During the last decades, dental implants have developed to be a widely accepted concept of rehabilitation after tooth loss. Analogous to natural teeth, implants are exposed to the intraoral biofilm. Therefore, they are also at risk to develop peri-implantitis, an inflammatory disease analogous to periodontitis at natural teeth. 

Supportive periodontal therapy (SPT) has widely been accepted as an essential continuation after successful treatment of periodontal diseases. SPT aims to prevent periodontal reinfection and, consequently, the recurrence of periodontitis. SPT has been proven to enormously contribute to the long-term preservation of periodontally compromised teeth [1,2,3]. Further tooth loss is described as a relatively rare event under sufficient SPT conditions [4,5,6,7,8,9,10,11]. 

Regrettably, the available literature indicates that patient compliance with SPT is generally insufficient [12,13,14]. Patients who had one or more dental implants inserted showed a significantly higher degree of compliance than patients without implants [15]. Nevertheless, it seems to be possible to generate a high level of patient compliance through improvements in patient communication and motivation at the end of active therapy [16].

Analogous to SPT, special oral hygiene measurements and treatment of implants are considered helpful to maintain the permanent health of peri-implant soft and hard tissues [17]. This condition is defined as the absence of inflammatory signs [18] and preserved soft and hard tissue dimensions [19]. In the case of peri-implant soft tissue inflammation, the homeostasis between microbial biofilm and the human host response in the peri-implant tissue has been lost [20]. The onset of peri-implant inflammation may be modulated by different risk factors and can lead to peri-implant mucositis, which is regarded by some investigators to be a precursor for peri-implantitis and, consequently, may cause implant loss [21,22].

Peri-implantitis has been defined as an inflammatory status of the peri-implant soft and hard tissue by pathological conditions causing progressive bone loss around the implant [22]. Poor plaque control has been described as a major risk factor for peri-implant diseases in several studies [23,24,25], whereas others do not see this correlation [26]. To achieve permanent peri-implant tissue health, special supportive implant therapy (SIT) programs have been developed to monitor and improve plaque control. Furthermore, patient communication should explain the importance of regular implant maintenance and motivate patients to continue treatment. A systematic review and meta-analysis summarized that systematic implant aftercare “seemed to reduce the rate of occurrence of peri-implant diseases” [27]. Nevertheless, the literature lacks data on patient compliance with SIT, especially in a private practice setting.

Therefore, the aim of this prospective study was to evaluate patient compliance rates with a systematic SIT program in private practice over a 10-year observational period and to assess possible impacting factors.

## 2. Material and Methods

The data of this prospective study were collected in a private dental practice with a focus on dental implants (Northern Hessia Implant Center). A prospective, noninterventional study design was used based on the analysis of primary patient data that had been extracted from patient records. We evaluated patient compliance rates with SIT over 10 years. A total of 233 patients with 524 implants were provided with implant-supported dentures in the study center and initially advised for implant maintenance and oral hygiene. Additionally, all patients were introduced to a structured postimplant maintenance program (SIT) with a 3-month recall. This study was authorized by the Ethics Commission of Albert-Ludwigs University Freiburg, Germany (No.: 261/19). Our study was conducted in compliance with STROBE guidelines. The study was internationally registered on 13 January 2019, in the German Clinical Trials Register (DRKS 00016570).

### 2.1. Study Population

All patients were identified who had been provided with dental implants and with implant-supported prostheses in the study center between 1 January 2005 and 31 December 2008. These patients were approached and were asked to participate in the study after having received written information regarding the aims and course of the present study. Patients who provided written informed consent and met the following inclusion criteria were included:-age ≥ 18 years;-receiving surgical and prosthetic implant therapy in the study center;-implant surgery received between January 2005 and December 2008; and-available data including anamnesis and clinical data during an observation period of at least 10 years.

The following criteria led to the exclusion of patients from the study:-taking medication that influences bone metabolism (i.e., bisphosphonates) or causes gingival hyperplasia;-having allergies against the materials used; and-exclusively referred for implant surgery.

The following implant systems were used: Ankylos (Dentsply Friadent, Mannheim, Germany), Branemark (Nobel Biocare, Gothenburg, Sweden), ITI Bonefit (Straumann, Freiburg, Germany), Astra Tech Implant System (Dentsply Friadent, Mannheim, Germany), and 3i (Biomet 3i, Karlsruhe, Germany).

### 2.2. Treatment

All patients were enlightened about the high relevance of postimplant maintenance for preventing peri-implant diseases at the beginning of the treatment, namely, when the treatment concept was presented. Therefore, every patient was informed that life-long SIT is an essential part of implant therapy.

All implants were placed either directly after the extraction of teeth that could not be preserved or after a minimum healing period of 3–4 months and according to the recommended protocol of the manufacturers. A structured prosthetic workflow for manufacturing the dentures was observed, and all dentures were provided by the same dental laboratory (BK Zahntechnik, Hofgeismar, Germany). All clinical procedures were provided by the same clinician (EF). After completion of the dentures, all patients were instructed in maintenance of the dentures, and, again, home-based oral hygiene with plaque control was explained. We recommended a 3-month recall for every patient. The goal of the long-term preservation of treatment effort and expense was pointed out. At this juncture, every patient received an appointment for the first SIT.

Compliance was defined in three grades and recorded yearly as follows:

Grade 0: no SIT appointments per year, no compliance;

Grade 1: one SIT appointment per year, medium compliance; and

Grade 2: two, three, or more appointments per year, high compliance.

If there were no 10-year data available, the patient was declined as dropout.

In every SIT session, oral hygiene instruction was repeated, and the teeth and implants were professionally cleaned. A detailed description of the proceedings during our SIT has been published before [28,29].

### 2.3. Data Analysis

All patients were prospectively evaluated according to the following general parameters:-age (at the time of implant insertion);-gender;-smoking habits;-medical history (heart disease, diabetes);-number of implants (after the examination time of 10 years);-implant position;-pre-existing experience in prophylactic programs;-geographical distance to the practice; and-surgical procedure of the implant insertion (immediate implant placement, sinus lift procedure)

Between 02.01.2019 and 11.30.2019, all patient files were examined, and the 10-year data were extracted. 

### 2.4. Statistical Analysis

The effect of different variables on compliance per year, the overall compliance rate for the whole period, and the influence of high or low compliance on clinical parameters were calculated by using robust regression analyses. To describe the data percentages, means and standard deviations were calculated. All calculations were performed with the statistical software STATA 16.1 (StataCorp LP, College Station, TX, USA). The level of significance was set at *p* < 0,05. 

This article focuses on the results of patient compliance and possible influencing factors. 

## 3. Results

### 3.1. Patient Characteristics

In our study, a total of 233 patients with 524 implants received implant-supported restorations between 2005 and 2008. Four patients (1.72%) with 10 implants did not have any appointments during the observation period. Therefore, we assessed 229 patients with 514 implants.

The observation time in all cases was 10 years. A total of 135 (59%) were female, and 94 (41%) were male. The average age of patients at the time of the implant treatment was 64.34 years. The patients’ pertinent data are provided in Table 1.

### 3.2. Implant Treatments

During the 10-year observational period, 12 (2.34%) implants had to be explanted; therefore, the survival rate was 97.66%. In total, 166 (32.3%) implants were inserted using the transalveolar sinus lift procedure, and in 92 (18%) cases, immediate implant placement was performed.

Of the 229 patients, 123 (53.71%) were provided with single crowns, 56 (24.45%) with bridges, 13 (5.68%) with overdentures, and 26 (11.35%) with a combination of fixed and removable dentures in one jaw. Eleven patients (4.80%) were provided with a combination of different denture types. Table 2 shows an overview of the different implant systems, and Table 3 provides the distribution of the quantity of implants per patient.

### 3.3. SIT Compliance

All patients were instructed initially about peri-implant hygiene and the importance of supportive implant therapy with 3-month recall intervals. The average number of SIT appointments was between 2.25 in the first year and 1.25 in year 10. For 87 (38%) patients, we had no 10-year data, and they declined as dropouts. Thus, 142 patients with 323 implants could be observed for 10 years.

During the first year after prosthetic deliverance, 5% did not comply and had no appointment within the first year. The compliance decreased annually, and in year 10, 38% of the patients had no SIT appointment, as shown in Figure 1**,** clarifying the average compliance every year.

There was no recognizably crucial timepoint at which most patients decided to continue SIT or to leave the program, but we observed that 66% of the patients, who showed one year of noncompliance, also had no compliance in the following year. Twenty-one percent of these patients returned to the SIT after one year with one appointment, and 12% had more than one appointment in the following year.

Seventeen patients (7.42%) showed total compliance with three or more appointments each year.

### 3.4. Analysis of Influence on Compliance

In the following, it will be examined which parameters had an influence on the annual compliance. Linear or logistic regression models were used for this purpose.

#### 3.4.1. Number of Implants Per Patient

The number of implants a patient received seemed to be a strong influential factor for compliance. Every year, patients with several implants had more appointments than patients with fewer or only one implant. During the first 3 years there was no significant difference to be noticed. From year 4 onwards we could observe a significantly higher compliance in patients with several implants (*p* < 0.05) compared to patients with few implants.

#### 3.4.2. Type of Implant-Supported Prostheses

Additionally, the type of denture the patients received seemed to have an influence on their compliance. Therefore, patients provided with prostheses showed the highest compliance over the whole examination period. This parameter showed significance in years 1 (*p* < 0.0001) and 4 (*p* = 0.039). In comparison, patients provided with bridges showed the lowest compliance in nearly every year. In years 1, 3, 5, 6, 7, and 10, this parameter reached significance (*p* < 0.05).

#### 3.4.3. Geographical Distance to Study Center

The third influential factor for patient compliance was the distance between the home and the study center. From year 4 onward, compliance decreased with increasing distance to the study center. This parameter could not reach significance.

#### 3.4.4. Cardiovascular Diseases and Smoking

Until year 7, patients with cardiovascular diseases had compliance comparable to that in patients without this parameter. In the last three years of observation, we could recognize a slightly lower compliance with significance in year 9 (*p* = 0.032). However, patients with a smoking habit showed better compliance until year 5, but the compliance decreased during year 7 and was lower than the compliance of patients without smoking habits during the last three years of observation. This parameter showed significance in year 9 (*p* = 0.047).

#### 3.4.5. Pre-Existing Experience in a Prophylaxis Program

Another seemingly influential factor was pre-existing experience in prophylaxis programs. Patients who had already been participating in a prophylaxis program before implantation showed recognizably better compliance than patients without this experience. From year 5 on, this parameter reached significance until year 10. Thus, we could detect the factor “pre-existing experience” seemed to be most crucial during the later years of the observational period. Figure 2 illustrate this correlation.

No correlation with patient compliance could be detected for “Diabetes”, “Age”, “Gender”, or “Increased surgical case complexity” (immediate implant placement, transalveolar sinus lift procedure).

#### 3.4.6. Reasons for Dropout

Of 87 patients, 19 (21.83%) passed away during the observational period, 2 (2.3%) moved away from the study center, 15 (17.24%) changed their dental provider, and 19 (21.83%) lost interest in the SIT program but remained in the study center. In 30 (34.48%) of the cases, there were other reasons for not remaining compliant until year 10 of the observational period.

## 4. Discussion

### 4.1. Main Results

This study revealed high rates of compliance, from 95% (year 1) to 60% (year 10), to a SIT program in a private practice over a 10-year observational period. No influential factors reached significance over the whole period. Weaker influences (significance in at least 1 of 10 years) were determined for the factors “Smoking”, “Pre-existing experience in prophylaxis programs”, “Number of implants”, “Type of denture”, and “Cardiovascular diseases.”

### 4.2. Limitations and Strengths

This study covered an observational period of 10 years, which is still rare in the literature, and can potentially increase the understanding of behaviors related to patient compliance. Additionally, there were too few patients to investigate all questions, so further research will be necessary. The fact that all treatments were performed by a single provider, in a single study center, and with a single study population might represent further limitations.

### 4.3. Interpretation

Implant treatment has been indicated to be a predictable option to provide different types of dentures to completely or partially edentulous patients. Many patients have decided in favor of this option.

The available data demonstrate great variation in the incidence of peri-implantitis [30], which can be explained by differences in study populations, observation periods of different lengths, variations in levels of maintenance measures, and different definitions of peri-implantitis [31]. Because there are no evidence-based guidelines for the treatment of peri-implantitis, prevention has become increasingly important, and several studies have demonstrated the positive influence of regular participation in a professional SIT program for the prevention of peri-implant inflammation [32,33,34,35,36,37]. Therefore, it is important to understand patient compliance behavior and possible influential factors.

Cardapoli and Gaveglio [15] concluded in 2012 that the insertion of an implant may have a psychological impact on patient compliance, as they recognized that patients with implants had a significantly higher compliance to SPT in an observation period of 5 years compared to patients without implants. Therefore, effort and expense would increase patient motivation. This outcome is supported by our results. The parameter “Number of implants” may be correlated with financial investment and time expenditure, and it was revealed to significantly increase our patient compliance with SIT, especially in the second half of the observation period.

We could recognize the type of denture as another strong influential factor. Patients with removable dentures showed the highest compliance with significance in years 1 and 4. This can be explained by the fact that, on the one hand, these patients have a larger number of implants, and, on the other hand, patients are more mindful of a removable prosthesis because they have to insert and remove it several times during the day for cleaning, sleeping, or eating. In contrast, patients with bridges showed the lowest compliance in nearly every year. One explanation could be that bridges are fixed dentures and are not as present in patients’ minds as removable dentures are. Therefore, it can be assumed that the need for maintenance is recognized to a lower extent. However, the question of why patients with bridges show lower compliance than patients with single crowns cannot be answered in this way. It can be assumed that patients with single crowns have more concern about their implants because they have chosen an implant after losing only one tooth, denying other treatment options, while patients with bridges have usually lost more than one tooth before deciding to receive an implant.

The geographical distance to the study center can hardly be compared to other studies due to considerable differences in local mobility, population density, and culture. Though we could recognize that patients with a very long distance to the study center (220 km) showed a very high compliance in the first 3 years, the compliance decreased from the year 4 with increasing distance. This result is comprehensible, as traveling longer distances is time consuming and cost intensive. A total of 60.3% of the patients in this study were already integrated into prophylaxis programs in the study center before implantation. They showed significantly better compliance than patients without this pre-existing experience in years 5, 7, 8, 9, and 10. This may be explained by the fact that the former were accustomed to having prophylaxis appointments and so did not lose motivation, especially in later years of observation.

Furthermore, we recognized that tobacco smokers had slightly better compliance in the first years of observation, but their compliance decreased from the fifth year onward, which can be explained by an initially higher motivation through contentious patient communications and a loss of motivation in further years. One reason for the decreased compliance could be a deteriorated health condition of these patients through secondary diseases caused by smoking. A similar explanatory approach could be applied to the group of patients with cardiovascular diseases. The noticeably decreased compliance in the last years of observation could be due to a worsened general condition.

## 5. Conclusions and Implications for Clinical Practice

Postimplant maintenance programs have been strongly recommended for the prevention of peri-implant disease. Risk factors such as diabetes and smoking are known as influential factors for the emergence and progression of inflammatory diseases of oral soft and hard tissue. Therefore, clinicians who introduce patients to SIT programs may aim for sufficient compliance, especially of patients with a higher risk for peri-implantitis, and will present SIT as an integral and indispensable part of implant therapy.

We observed that patient motivation for regular SIT participation decreased significantly over the 10 years of observation. None of the abovementioned parameters showed statistical significance over the entire period, but we can highlight that the number of implants, the type of denture, pre-existing experience in prophylaxis programs, and the geographical distance to the study center seem to be the strongest influences.

## Figures and Tables

**Figure 1 jcm-09-01988-f001:**
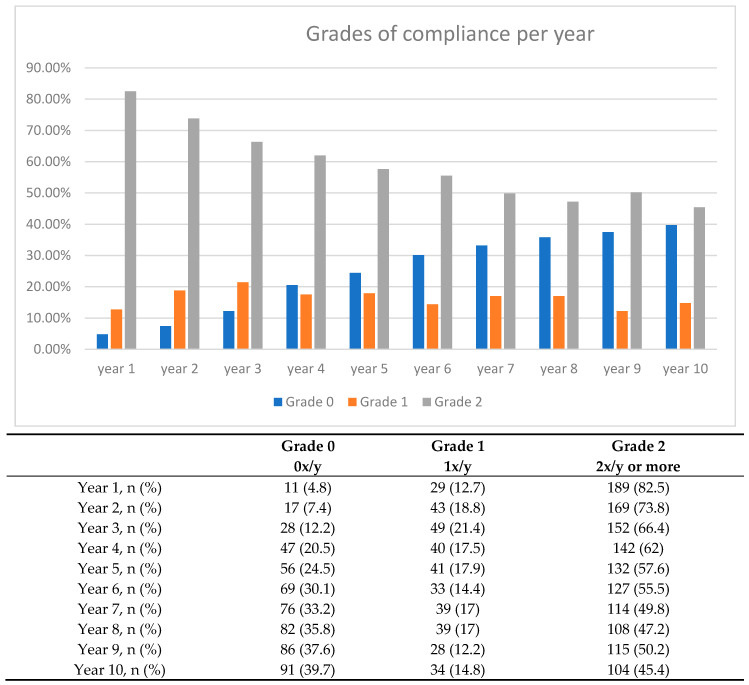
Overview of the grades for compliance according to each year.

**Figure 2 jcm-09-01988-f002:**
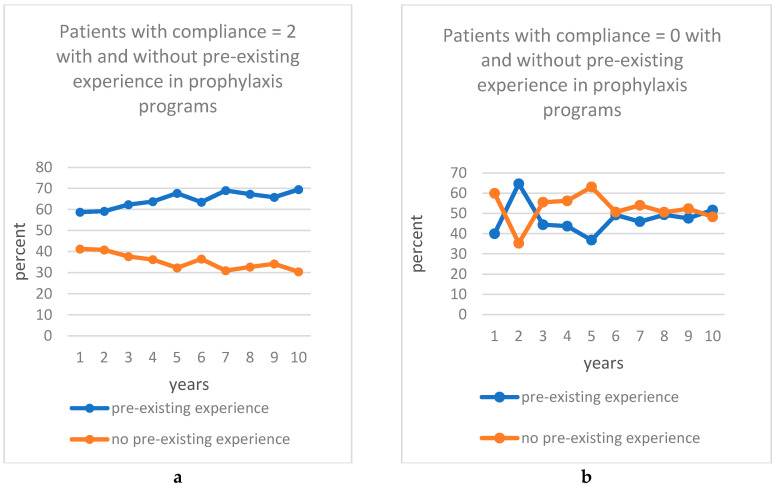
Depiction of the compliance depending on the pre-existing experience in prophylaxis programs; (**a**) Percentage of patients with compliance = 2 with and without pre-existing experience in prophylaxis programs from year 0 to year 10; (**b**) Percentage of patients with compliance = 0 with and without pre-existing experience in prophylaxis programs from year 0 to year 10.

**Table 1 jcm-09-01988-t001:** Pertinent patient data.

Observational Period		10 Years	
Gender	Male	94	41%
	Female	135	59%
Illnesses	Diabetes mellitus	8	3.5%
	Cardiovascular diseases	68	29.7%
Active smoker		16	7%
Average age in years		64.34	
Experience with the prophylaxis program	Yes	138	60.3%
	No	91	39.7%

**Table 2 jcm-09-01988-t002:** Implant systems.

	Number of Implants	Percentage
Ankylos	416	81
Branemark	68	13
ITI Bonefit	1	0.2
Astra	25	5
3i	4	0.8
Total	514	100

**Table 3 jcm-09-01988-t003:** Number of implants per patient.

Number of Implants	Frequency	Percent
1	72	31.4
2	68	29.7
3	20	8.7
4	31	13.5
5	9	3.9
6	10	4.4
7	4	1.7
8	6	2.6
9	4	1.7
10	2	0.9
11	1	0.4
12	1	0.4
16	1	0.4
Total	229	99.7 *

* Difference up to 100% is due to rounding effects.

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
