# Peer review of "Supportive Implant Therapy (SIT): A Prospective 10-Year Study of Patient Compliance Rates and Impacting Factors"

_jcm, 2020, doi:10.3390/jcm9061988_

Round 1

Reviewer 1 Report

The present paper evaluates how “supportive periodontal therapy” can be arranged in a group of implant patients. There is no control group in the present paper or in most of their quoted papers. The major problem with this paper is that the authors too uncritically draw most positive conclusions about their SIT approach, so positive effects that they “should be pointed out repeatedly” without realizing that others see similarly good long term clinical results without using the SIT approach. The possible common denominator between those using SIT and those who do not may be that they are all quite experienced and good clinicians, perhaps the most important factor of all behind a positive clinical outcome.

The first three paragraphs in the introduction are related mainly to teeth and periodontitis. This is largely irrelevant for an implant paper. This part has to be shortened and most of the 21 references in this part of the paper must be deleted since they refer to teeth and not implants. In the revised and clearly shortened part of the introduction please see to that all statements presented about periodontal matters are backed up by properly controlled data.

The SIT approach results in 97,6 % of implant survival at 10 years, if with a very high implant unaccounted for rate of 38%. To the uncritical reader this high survival rate may be interpreted as evidence that only the SIT approach presents such positive results. However, similar positive results are obviously possible without the numerous recalls of a life-long SIT. Therefore, the authors must quote other papers too for scientific balance. Wennerberg et al Eur J Oral IMplantology suppl 2018 described 10 year clinical results of, among other designs, moderately rough implant brands that were not followed up in the meticulous SIT manner described in the present paper. The 10 year survival outcome of moderately rough Astra, Straumann and Nobel implants were between 95 and 99%, i.e. most similar survival data as those reported in the present paper. These alternative 10 year data were only retrospective as a drawback, but they saw much less drop-out implants cf the present SIT-study as a particular advantage. The Wennerberg et al data must be quoted for balance. Based on the Wennerberg et al data the authors of the present paper will have to delete all statements that perio maintenance programs are needed for good clinical results. Delete the first sentence in the Conclusions.

Conclusions need to be changed since this paper has failed in demonstrating that patients really must be included in profylaxis programs, since similar long term results as in the present study has been matched by other studies where the profylaxis program has not been used. In uncontrolled studies such as the present one, it is imperative to avoid drawing too strong conclusions.

Line 55: This strong writing with respect to hygiene measures etc would put some demand on that reference 18 is a controlled study where particular cleaning has been compared to other approaches without cleaning. In case reference 18 is not a controlled study, this has to be pointed out. That we all prefer clean patients is something else.

Lines 59 to 60 …..and can lead to peri-implant mucositis which known to be a precursor for peri-implantitis… This is too strong. Donath et al(1992) reported that all functioning implants had a chronic inflammation. Instead of “known to be” write: “regarded by some investigators to be…..” with references 22, 23 kept.

Line 63 “Poor plaque control …..a major risk for peri-implant diseases”. Here the authors must quote the Menini et al paper published in Int J Prosthodontics of 2018, p337, for balance. These authors found no correlation between plaque and marginal bone loss in a clinical paper followed up over 14 years. In a second study published in J of Clinical Medicine 2019 the authors presented an explanation to these findings. I certainly agree that the authors may quote references 24-26 too but the Menini et al work cannot be ignored. Some investigators believe poor plaque control to present a major risk for peri-implantitis whereas others do not see this correlation..

Line 68. I severely doubt this statement from reference 27. In the abstract the authors of paper 27 write “Supportive periodontal therapy seemed to reduce the rate of occurrence of peri-implant diseases”, with other words a much less strong observation than the one presented by the present authors. Admittedly, due to the Corona situation I am not at our laboratories where I have the full version of this paper. I suggest the present authors change to quoting the writing in the abstract and avoid the 11-fold increase. If not, I will ask them to summarize a great number of long term papers where SIT was not used and the frequency of peri-implantitis nevertheless was minor.

Wheras the Materials and Methods section include information about results from various types of probing and of evaluations of marginal bone loss, no data from these investigations are presented in the paper. The reason for this is unknown and must be reported.

The sentence on line 263 must be deleted.

Reference 11 is incomplete.

Author Response

The present paper evaluates how “supportive periodontal therapy” can be arranged in a group of implant patients. There is no control group in the present paper or in most of their quoted papers. The major problem with this paper is that the authors too uncritically draw most positive conclusions about their SIT approach, so positive effects that they “should be pointed out repeatedly” without realizing that others see similarly good long term clinical results without using the SIT approach.

  • Thank you for this relevant point. Maybe we caused a misunderstanding: The term ‚Supportive Periodontal Therapy -SPT‘ has been accepted worldwide for different existing aftercare programs following systematic periodontal therapy. Implant aftercare connot be subsumed under this term because ‚periodontal‘ implicates the existence of at least one tooth with a periodontium. In many cases, implant patients are edentulous. Therefore, some years ago we proposed an analoguous term ‚Supportive Implant Therapy – SIT‘ for systematic implant aftercare programs. This did not aim exclusively at our special program, but at the different existing programs of systematic implant aftercare. In the meantime, other researchers started to use this term too. Therefore, we stated in the ‚Introduction‘ section (Line 64/65):

„To achieve permanent peri-implant tissue health, special supportive implant therapy (SIT) programs have been developed to monitor and improve plaque control”

So, we absulutely did not want to claim that our aftercare system would be superior to others. All those are SIT. Moreover, we did not intend to deny that one may reach very good long-term results without any aftercare system.  In this paper, we want to present data of patients‘ compliance with our concept of systematic SIT.

We apologize for having been imprecise.

The possible common denominator between those using SIT and those who do not may be that they are all quite experienced and good clinicians, perhaps the most important factor of all behind a positive clinical outcome.

  • We absolutely agree with this statement

The first three paragraphs in the introduction are related mainly to teeth and periodontitis. This is largely irrelevant for an implant paper. This part has to be shortened and most of the 21 references in this part of the paper must be deleted since they refer to teeth and not implants. In the revised and clearly shortened part of the introduction please see to that all statements presented about periodontal matters are backed up by properly controlled data.

  • With all due respect, we can agree only partially. Peri-implantitis and periodontitis are very analougous phenomenons. This concerns etiology (pathogeneity of intraoral biofilms), therapy and prevention. Until today, peri-implant therapy concepts are largely orientated to systematic perio therapy. Both systematic aftercare programs aim at controlling the biofilm and therefore, they aim at the prevention of inflammation, tissue destruction and -finally- loss. While in literature, data on the relevant issue of this paper -patients compliance with SIT- have widely been lacking, it might be helpful for readers to develop what is known about SPT compliance as a reference. So we cannot consider this part completely irrelevent. Nevertheless we follow the arguments of this reviewer as far as possible and shortened this part considerably. Moreover, we tried to focus on the rationale of this study:

1. Introduction

During the last decades, dental implants have been developed to a widely accepted concept of rehabilitation after tooth loss. Analouguous to natural teeth, implants are exposed to the intraoral biofilm. Therefore, they are also at risk to develop peri-implantitis, an inflammatory disease analoguous to periodontitis at natural teeth.

  Supportive periodontal therapy (SPT) has widely been accepted as an essential continuation after successful treatment of periodontal diseases. SPT aims to prevent periodontal reinfection and, consequently, the recurrence of periodontitis. SPT has been proven to enormously contribute to the long-term preservation of periodontally compromised teeth [1–3]. Further tooth loss is described as a relatively rare event under sufficient SPT conditions [4–11].

Regrettably, the available literature indicates that patient compliance with SPT is generally insufficient [12, 13, 14]. Patients who had one or more dental implants inserted showed a significantly higher degree of compliance than patients without implants [15]. Nevertheless, it seems to be possible to generate a high level of patient compliance through improvements in patient communication and motivation at the end of active therapy [16].

Analogous to SPT, special oral hygiene measurements and treatment of implants are considered helpful to maintain the permanent health of peri-implant soft and hard tissues [17]. This condition is defined as the absence of inflammatory signs [18] and preserved soft and hard tissue dimensions [19]. In the case of peri-implant soft tissue inflammation, the homeostasis between microbial biofilm and the human host response in the peri-implant tissue has been lost [20]. The onset of peri-implant inflammation may be modulated by different risk factors and can lead to peri-implant mucositis, which is regarded by some investigators to be a precursor for peri-implantitis and consequently may cause implant loss [21, 22].

Peri-implantitis has been defined as an inflammatory status of the peri-implant soft and hard tissue by pathological conditions causing progressive bone loss around the implant [22]. Poor plaque control has been described as a major risk factor for peri-implant diseases in several studies [23–25], whereas others do not see this correlation [26]. To achieve permanent peri-implant tissue health, special supportive implant therapy (SIT) programs have been developed to monitor and improve plaque control. Furthermore, patient communication should explain the importance of regular implant maintenance and motivate patients to continue treatment. A systematic review and meta‐analysis summarized that systematic implant aftercare ‘seemed to reduce the rate of occurrence of peri-implant diseases‘ [27]. Nevertheless, the literature lacks data on patients’ compliance with SIT -especially in a private practice setting.

Therefore, the aim of this prospective study was to evaluate patient compliance rates with a systematic SIT program in private practice over a 10-year observational period and to assess possibly impacting factors.

The SIT approach results in 97,6 % of implant survival at 10 years, if with a very high implant unaccounted for rate of 38%. To the uncritical reader this high survival rate may be interpreted as evidence that only the SIT approach presents such positive results. However, similar positive results are obviously possible without the numerous recalls of a life-long SIT. Therefore, the authors must quote other papers too for scientific balance. Wennerberg et al Eur J Oral IMplantology suppl 2018 described 10 year clinical results of, among other designs, moderately rough implant brands that were not followed up in the meticulous SIT manner described in the present paper. The 10 year survival outcome of moderately rough Astra, Straumann and Nobel implants were between 95 and 99%, i.e. most similar survival data as those reported in the present paper. These alternative 10 year data were only retrospective as a drawback, but they saw much less drop-out implants cf the present SIT-study as a particular advantage. The Wennerberg et al data must be quoted for balance. Based on the Wennerberg et al data the authors of the present paper will have to delete all statements that perio maintenance programs are needed for good clinical results. Delete the first sentence in the Conclusions.

  • We apologize again for obviously having caused a basic misunderstanding. This study does not assess 10 years survival and success rates of dental implants. The last sentence of our ‚Introduction‘ section -focusing the rationale of this study- was:

Therefore, the aim of this prospective study was to evaluate patient compliance rates with a systematic SIT program in private practice over a 10-year observational period.“

                Moreover, the title is:

Supportive Implant Therapy (SIT): Patient Compliance Rates and Impacting Factors. A Prospective 10-Year Study”

We do not intend to present or to discuss 10 years survival and succes rates of dental implants. To make this fact more clear and to respond to the criticism of this reviewer, we

  1. Rephrased in the M+ M section (Line 77):

„We evaluated the patients’ compliance rates with SIT over 10 years.”

  1. We omitted in the M+M section (Line 139 – 144)

This examination also included many clinical parameters to determine the presence of inflammation (PBI, BOP, suppuration) and peri-implant tissue loss (pocket probing depth) as technical complications of the implants and dentures (fractures, decementation, screw or abutment loosening, screw breakage, adhesion loss with the need for relining or the exchange of holding units).

The clinical parameters were taken using a millimeter-scale periodontal probe (PCP 15, Hu-Friedy, Chicago, II, USA), and the PPD was measured at four measurement points per implant.

  1. In the ‚Resuts‘ section we omitted (Line 176/177):

During the 10-year observational period, 12 (2.34%) implants had to be explanted; therefore, the survival rate was 97.66%.

  1. In the ‚Discussion‘ section we omitted (Line 311):

„strongly“

As different rersearchers do recommend postimplant maintenance programs for the prevention of peri-implant disease, we are not able to omit the entire sentence.

Conclusions need to be changed since this paper has failed in demonstrating that patients really must be included in profylaxis programs, since similar long term results as in the present study has been matched by other studies where the profylaxis program has not been used. In uncontrolled studies such as the present one, it is imperative to avoid drawing too strong conclusions.

  • We agree and omitted in the ‚Conclusion‘ section (Line 321 – 324)

We can presume that patients should be introduced into prophylaxis programs as soon as possible, that the benefits of SIT (long-term preservation of treatment effort and expense) should be pointed out repeatedly, and that the first SIT appointment should be made during the first prosthesis delivery session.

Line 55: This strong writing with respect to hygiene measures etc would put some demand on that reference 18 is a controlled study where particular cleaning has been compared to other approaches without cleaning. In case reference 18 is not a controlled study, this has to be pointed out. That we all prefer clean patients is something else.

  • We agree and replaced the term „necessary“ with the term „helpful“:

Lines 59 to 60 …..and can lead to peri-implant mucositis which known to be a precursor for peri-implantitis… This is too strong. Donath et al(1992) reported that all functioning implants had a chronic inflammation. Instead of “known to be” write: “regarded by some investigators to be…..” with references 22, 23 kept.

  • We agree and replaced the term „known“ by th term „regarded by some investigators to be..“

Line 63 “Poor plaque control …..a major risk for peri-implant diseases”. Here the authors must quote the Menini et al paper published in Int J Prosthodontics of 2018, p337, for balance. These authors found no correlation between plaque and marginal bone loss in a clinical paper followed up over 14 years. In a second study published in J of Clinical Medicine 2019 the authors presented an explanation to these findings. I certainly agree that the authors may quote references 24-26 too but the Menini et al work cannot be ignored. Some investigators believe poor plaque control to present a major risk for peri-implantitis whereas others do not see this correlation..

  • Thank you very much for this helpful advice. We agree and added in the ‚Introduction‘ section (Line 85):

“Poor plaque control has been described as a major risk factor for peri-implant diseases in several studies [23–25], whereas others do not see this correlation [26].“

Line 68. I severely doubt this statement from reference 27. In the abstract the authors of paper 27 write “Supportive periodontal therapy seemed to reduce the rate of occurrence of peri-implant diseases”, with other words a much less strong observation than the one presented by the present authors. Admittedly, due to the Corona situation I am not at our laboratories where I have the full version of this paper. I suggest the present authors change to quoting the writing in the abstract and avoid the 11-fold increase. If not, I will ask them to summarize a great number of long term papers where SIT was not used and the frequency of peri-implantitis nevertheless was minor.

  • We doublechecked this point and we agree. We apologize for this shortcoming. The 11fold elevated risk seemes to origin from only one of the included studies (Rinke S, Ohl S, Ziebolz D, Lange K, Eickholz P. Prevalence of periimplant disease in partially edentulous patients: A practice-based cross-sectional study. Clin Oral Implants Res 2011;22:826-833.) but was not part of the cited Atieh paper (Ref 27).

Therefore, we thank you very much for this correction and rephrased in the ‚Introduction‘ section (Line 66/67):

A systematic review and meta‐analysis summarized that systematic implant aftercare ‘seemed to reduce the rate of occurrence of peri-implant diseases‘ [27].

Wheras the Materials and Methods section include information about results from various types of probing and of evaluations of marginal bone loss, no data from these investigations are presented in the paper. The reason for this is unknown and must be reported.

  • Thank you for this point. We consequently omitted this content (See above).

The sentence on line 263 must be deleted.

  • We agree and omitted this sentence.

Reference 11 is incomplete.

  • We apologize for this shortcoming and completed reference 11

Submission Date               17 May 2020

Date of this review          20 May 2020 08:05:08

Fínally, we want to thank all reviewers for their work on our paper which led to significant improvements and more clarification. We hope that we were able to adress all concerns of the reviewers in a satisfactory manner.

For all authors

Dr. Eberhard Frisch

Reviewer 2 Report

Comments to the authors

I have read the manuscript “Supportive Implant Therapy (SIT): Patient Compliance Rates and Impacting Factors. A Prospective 10-Year Study”. The aim of this prospective study was to evaluate patient compliance rates with a 71 systematic SIT program in private practice over a 10-year observational period.

I find all the parts of the paper properly developed therefore I think the manuscript could be published on the Journal of Clinical Medicine, but I think the authors should made some modifications.

 Abstract

I could not find in the abstract, which is the main objective of the study. The main objective is not clearly defined.

Introduction

The authors wrote: “patients who had one or more dental implants inserted showed a significantly higher degree of compliance than patients without implants”: Why do they say that? Can the authors attach a reference to support that affirmation or it is just an opinion?

The authors say: "A prospective noninterventional study design was used based on the analysis of primary patient data that had been extracted from the patients’ records. We evaluated the clinical and radiological data of implants after 10 years ". If the authors have evaluated the data "after 10 years" through the records, the study is not prospective but retrospective. If the authors designed the study 10 years ago, and they have collected the information during 10 years following the patients evolution, then is prospective. The key is, which is the position of the researcher in the time line?

Material and methods

Equator indicates which guidelines the authors should use, but EQUATOR is not a guideline (line 82).

Results

Table 4: years 3 and 5 do not sum 100%.

Discussion

The authors should improve this section properly.  The authors have written a discussion section with few references.

"Type of denture", "Geographical distance" or "pre-existence experience" have the authors found some references related to this parts of the discussion?

The authors wrote: “we recognized that tobacco smokers had slightly better compliance in the first years of observation”: could the author compare this section witd dome others studies?

Regarding the lack of this kind of study, which is de ideal design of study that should be developed?

Funding

The authors do no say anything in this part, but explain the funding in the aknowledgements.

References

From references 1 to 30, the authors used Vancouver, but from 31 to 37, they mixed Vancouver with Harvard, placing the year after the authors.

Author Response

I have read the manuscript “Supportive Implant Therapy (SIT): Patient Compliance Rates and Impacting Factors. A Prospective 10-Year Study”. The aim of this prospective study was to evaluate patient compliance rates with a 71 systematic SIT program in private practice over a 10-year observational period.

I find all the parts of the paper properly developed therefore I think the manuscript could be published on the Journal of Clinical Medicine, but I think the authors should made some modifications.

  • Thank you very much for this friendly words

 Abstract

I could not find in the abstract, which is the main objective of the study. The main objective is not clearly defined.

  • Thank you very much for this clarification. We apologize for this shortcoming and rephrased in Line 14 – 16:

“Abstract: The main objective of this study is to present patient compliance rates and influential factors for regular attendance in a systematic implant aftercare program (Supportive Implant Therapy -SIT) within a 10-year observation period.”

Introduction

The authors wrote: “patients who had one or more dental implants inserted showed a significantly higher degree of compliance than patients without implants”: Why do they say that? Can the authors attach a reference to support that affirmation or it is just an opinion?

  • Thank you very much for this clarification. This conclusion is drawn still related to citation No 16. To make this clear, we added [16] in Line 55 / now Line 76

The authors say: "A prospective noninterventional study design was used based on the analysis of primary patient data that had been extracted from the patients’ records. We evaluated the clinical and radiological data of implants after 10 years ". If the authors have evaluated the data "after 10 years" through the records, the study is not prospective but retrospective. If the authors designed the study 10 years ago, and they have collected the information during 10 years following the patients evolution, then is prospective. The key is, which is the position of the researcher in the time line?

  • Thank you for this point. We discussed this point too. In fact, this study was designed any years ago. We collected the data permanently during the 10 years observatinal period and started the evaluation at the end of this period. Therefore, we decidid for ‚prospective‘. Of course, if we are requested, we will change to ‚retrospective‘

Material and methods

Equator indicates which guidelines the authors should use, but EQUATOR is not a guideline (line 82).

  • We apologize for having been imprecise and rephrased in the M + M section (Line 104):

„Our study was conducted in compliance with STROBE guidelines.”

Results

Table 4: years 3 and 5 do not sum 100%.

  • We doublechecked the tables and agree. We apologize for having been imprecise. In Table 3, we corrected 2 values, added 7 in the last line and put a * with the relevent explanation:

*Difference up to 100% is due to rounding effects

In Table 4, we corrected 2 values.

Discussion

The authors should improve this section properly.  The authors have written a discussion section with few references.

"Type of denture", "Geographical distance" or "pre-existence experience" have the authors found some references related to this parts of the discussion?

  • Thank you very much for mentioning these relevent question. Unfortunetely, tot he best of our knowledge, ralevant data on these possible impact factors have not been published in literature.

The authors wrote: “we recognized that tobacco smokers had slightly better compliance in the first years of observation”: could the author compare this section with some others studies?

  • Thank you very much for this point. We also would like to do so, but to the best of our knowledge, no relevant data has been available.

Regarding the lack of this kind of study, which is the ideal design of study that should be developed?

  • Thank you very much for this relevant question. We think that the lack of data until today might be caused by the fact that it is very time-consuming and very extensive to conduct suchlike long-term studies on compliance rates. Furthermore, systematic aftercare programs have not been introduced yet in all implant centers -unfortunately. The longer the observational time period will be, the more dropouts will be countes. Usually it is very extensive to follow these.

So we tried to follow an appropiate study design – as good as we could.

Funding

The authors do no say anything in this part, but explain the funding in the aknowledgements.

  • Thank you for this advice, but to the best of our knowledge, we do not have a ‚acknowledge‘ section. The ‚Funding‘ statement was placed after the end of the ‚conclusion’ section between ‚Author contribution‘ and ‚Conflict of Interest‘ statements. To the best of our knowledge, this should be the right place.

References

From references 1 to 30, the authors used Vancouver, but from 31 to 37, they mixed Vancouver with Harvard, placing the year after the authors.

  • We apologize for this shortcoming and corrected the relevant references

Submission Date               17 May 2020

Date of this review          25 May 2020 10:24:47

Fínally, we want to thank all reviewers for their work on our paper which led to significant improvements and more clarification. We hope that we were able to adress all concerns of the reviewers in a satisfactory manner.

For all authors

Dr. Eberhard Frisch

Reviewer 3 Report

Reviewer Comments to Author: 
Dear authors, 
I appreciate the efforts of the study with the title "
Supportive Implant Therapy (SIT): Patient Compliance Rates and Impacting Factors. A Prospective 10-Year Study". In principle, the topic of this investigation is of potential clinical interest.

Design:

10 Years follow up is a very good time and the study has a sufficient sample size. Although the study may have had a prospective approach of the evaluation, the data was collected from existing records and therefore it is a retrospective study in my opinion.

Has the SIT had an influence on the survival rate?

Authors stated that radiographs were evaluated. Furthermore, “this examination also included many clinical parameters to determine the presence of inflammation (PBI, BOP, suppuration) and peri-implant tissue loss (pocket probing depth) as technical complications of the implants and dentures (fractures, decementation, screw or abutment loosening, screw breakage, adhesion loss with the need for relining or the exchange of holding units). The clinical parameters were taken using a millimeter-scale periodontal probe (PCP 15, Hu- Friedy, Chicago, II, USA), and the PPD was measured at four measurement points per implant.“

” We evaluated whether or when an implant received the diagnosis of peri-implant mucositis and peri-implantitis, how the peri-implantitis was treated (chirurgical with hard and/or soft tissue augmentation or nonchirurgical by frequent peri-implant cleaning for bacteria reduction) and how many patients exhibited peri-implantitis at the point of exactly 10 years after implant insertion. Additionally, radiographs of the implants taken 10 years after implant insertion, using the parallel technique, were analyzed. The radiographic linear distance from the implant shoulder to the first bone-to-implant contact was used to calculate the marginal bone levels.”

However, the results for the mentioned assessments are not presented. Authors state that this article focuses on the results of patient compliance and possible influencing factors. Are the results published in another manuscript? This reviewer feels that the mentioned results should be presented as this is very interesting. For example, has the SIT had an influence on these parameters, such as periimplantitis, bone loss, …

Maybe the authors plan to publish these results I a separate manuscript. In this case the cited sections above need to be removed and as a consequence of this, the value of this manuscript strongly decreases.

Recommendations:

The presented parameters that have had an influence on the annual compliance show the reader what patient group needs to be motivated during SIT. However, many substantial parameters that were evaluated were not presented.

I recommend to rewrite the manuscript and add more data (e.g success rate, bone loss, clinical cases, …). Furthermore, the statistical analysis should also evaluate SIT on these parameters. This approach could be used to investigate the influence of SIT on implant parameters, which would be very important for clinicians and patients.

Author Response

I appreciate the efforts of the study with the title "Supportive Implant Therapy (SIT): Patient Compliance Rates and Impacting Factors. A Prospective 10-Year Study". In principle, the topic of this investigation is of potential clinical interest.

  • Thank you very much for this valuation. 

Design:

10 Years follow up is a very good time and the study has a sufficient sample size. Although the study may have had a prospective approach of the evaluation, the data was collected from existing records and therefore it is a retrospective study in my opinion.

  • Thank you for mentioning this point. We discussed this topic too. In fact, this study was designed many years ago. We collected the data permanently during the 10 years observational period and started the evaluation at the end of this period. Therefore, we decided for ‚prospective‘. Of course, if we are requested, we will change to ‚retrospective‘

Has the SIT had an influence on the survival rate?

  • Thank you very much for this question. Unfortunately, this question was not assessed in the

present study. As we are interested in this point too, we conducted a different study with a totally different clientele evaluating this topic. It was published recently (Frisch, E.; Vach, K.; Ratka‐Krueger, P. Impact of supportive implant therapy on peri‐implant diseases: A retrospective 7‐year study. J. Clin. Periodontol. 2019, 47, 101–109) and revealed a ~ 4fould elevated risk for peri-implantitis among the SIT noncompliant group of patients.

Authors stated that radiographs were evaluated. Furthermore, “this examination also included many clinical parameters to determine the presence of inflammation (PBI, BOP, suppuration) and peri-implant tissue loss (pocket probing depth) as technical complications of the implants and dentures (fractures, decementation, screw or abutment loosening, screw breakage, adhesion loss with the need for relining or the exchange of holding units). The clinical parameters were taken using a millimeter-scale periodontal probe (PCP 15, Hu- Friedy, Chicago, II, USA), and the PPD was measured at four measurement points per implant.“

” We evaluated whether or when an implant received the diagnosis of peri-implant mucositis and peri-implantitis, how the peri-implantitis was treated (chirurgical with hard and/or soft tissue augmentation or nonchirurgical by frequent peri-implant cleaning for bacteria reduction) and how many patients exhibited peri-implantitis at the point of exactly 10 years after implant insertion. Additionally, radiographs of the implants taken 10 years after implant insertion, using the parallel technique, were analyzed. The radiographic linear distance from the implant shoulder to the first bone-to-implant contact was used to calculate the marginal bone levels.”

However, the results for the mentioned assessments are not presented. Authors state that this article focuses on the results of patient compliance and possible influencing factors. Are the results published in another manuscript? This reviewer feels that the mentioned results should be presented as this is very interesting. For example, has the SIT had an influence on these parameters, such as periimplantitis, bone loss, …

Maybe the authors plan to publish these results I a separate manuscript. In this case the cited sections above need to be removed

  • Thank you very much for this clarification. As stated above, this study aims to present data on patients’ comliance rates with an SIT program over a 10 year observation period and to assess possibly impacting factors.

We apologize for these misleading sentences and removed them from the manuscript. The M + M section ended and still ends with the sentence:

This article focuses on the results of patient compliance and possible influencing factors.”

and as a consequence of this, the value of this manuscript strongly decreases.

  • With all due respect, we regret not to be able to agree completely. We repeatedly stated (Title, Abstract, Introduction, M+M) that the present study assesses SIT compliance and possibly impacting factors. In this field, until today we must assert widely lacking data -especially concerning long-term observational periods ≥ 10 years and originating from a private practice setting. In this paper, we try to present suchlike data. We did not claim to present different content.

Therefore, we cannot see in which way the present manuscript would decrease in value.

Recommendations:

The presented parameters that have had an influence on the annual compliance show the reader what patient group needs to be motivated during SIT. However, many substantial parameters that were evaluated were not presented.

I recommend to rewrite the manuscript and add more data (e.g success rate, bone loss, clinical cases, …). Furthermore, the statistical analysis should also evaluate SIT on these parameters. This approach could be used to investigate the influence of SIT on implant parameters, which would be very important for clinicians and patients.

  • Thank you very much for this appreciation and for the recommendation. We agree. Therefore and once again we refer to the abovementioned study which evatuated this special topic.

Submission Date               17 May 2020

Date of this review          23 May 2020 23:25:47

Fínally, we want to thank all reviewers for their work on our paper which led to significant improvements and more clarification. We hope that we were able to adress all concerns of the reviewers in a satisfactory manner.

For all authors

Dr. Eberhard Frisch

Round 2

Reviewer 1 Report

This paper is now clearly improved. The authors have understood the limitations with respect to strong conclusions inherit in a non-controlled study. The text is more balanced. I recommend the paper for publication.

Author Response

Thank you very much for your positive votum! We are happy having adressed all points you raised in your valuable review.

Again, we want to thank all reviewers for their time and their efforts to increase the quality of our study in a considerable degree.

For all authors

Dr. Eberhard Frisch

Reviewer 2 Report

I have reviewed again the manuscript “Supportive Implant Therapy (SIT): Patient Compliance Rates and Impacting Factors. A Prospective 10-Year Study.”

The authors have addressed and revised their manuscript according to my former comments.

As I stated in the last revision, I find all the parts of the paper properly developed therefore I think the manuscript could be published on the Journal of Clinical Medicine.

Kind regards

Author Response

(The authors gave the same response as above.)

Reviewer 3 Report

Dear authors, 
I appreciate the efforts of the revision with the title "Supportive Implant Therapy (SIT): Patient Compliance Rates and Impacting Factors. A Prospective 10-Year Study".

I regret to inform you that I recommend against publishing your manuscript, although the topic is of potential interest. According to the high number of recommendations that have been raised, the manuscript required major improvements. Unfortunately, the manuscript was not sufficiently rewritten. The authors did not completely explain, why the first mentioned assessments were not presented/added. Furthermore, additional data would have improved the manuscript but was not sufficiently added, such as survival rate analysis, success rate, e.g. clinical cases. Dealing with dental implant research these standard parameters should be assessed if possible.

Author Response

Reviewer 3

Review Report Form

Open Review

English language and style

( ) Extensive editing of English language and style required
( ) Moderate English changes required
( ) English language and style are fine/minor spell check required
(x) I don't feel qualified to judge about the English language and style

Yes

Can be improved

Must be improved

Not applicable

Does the introduction provide sufficient background and include all relevant references?

( )

( )

(x)

( )

Is the research design appropriate?

( )

(x)

( )

( )

Are the methods adequately described?

( )

(x)

( )

( )

Are the results clearly presented?

( )

( )

(x)

( )

Are the conclusions supported by the results?

( )

(x)

( )

( )

Comments and Suggestions for Authors

Dear authors, 

I appreciate the efforts of the revision with the title "Supportive Implant Therapy (SIT): Patient Compliance Rates and Impacting Factors. A Prospective 10-Year Study".

Thank you very much for this appreciation.

I regret to inform you that I recommend against publishing your manuscript, although the topic is of potential interest.

Thank you very much for your positive judgement concerning SIT compliance data.

According to the high number of recommendations that have been raised, the manuscript required major improvements.

We agreed and in consequence, we adressed all recommendations of all reviewers point by point.

Unfortunately, the manuscript was not sufficiently rewritten.

We are very concerned about this severe judgement. All topics raised by three reviewers were adressed intensively item-to-item. In consequence, two reviewers are absolutely satisfied and recommend this paper unrestrictedly for publication in the JCM.

Moreover, all recommendations of this reviewer (excluded one single that was substantiated) should have been adressed sufficiently as well, because no further comments were raised. With all due respect, this formulation in a perceived slightly degrading manner might not be justified so far.

The authors did not completely explain, why the first mentioned assessments were not presented/added.

If so perceived, we apologize for not having been clear enough. We want to correct this shortcoming. Unfortunately, we are not able to present data concerning clinical values on the included implants, because suchlike data have not been available. As we did not intend to conduct this assessment on peri-implant clinical data but exclusively on data concerning the rationale of this paper: SIT compliance rates and possibly impacting factors. In a preceding publication [Ref No 29 J Clin Periodontol 2014], an analogous study design was used and accepted without any restriction.

We regret that due to an error on our side it might have seemed that clinical data would have been involved. Therefore and following your advice, we removed the misleading lines.

Furthermore, additional data would have improved the manuscript but was not sufficiently added, such as survival rate analysis, success rate, e.g. clinical cases.

Of course we agree. Many published studies would have been improved if more data or additional data or different data would have been collected and included. But often it is not possible to do what was desirable. The abovementioned desired data describes a significantly different type of study which was not intended and aimed at. We cannot add data we don’t have. These data might have been of maybe some interest, but typical implant survival and success studies covering observational periods up to 10 years have been published many times before. Therefore, we are actually working on two suchlike clinical studies covering observational periods of 20 and 25 years.

Dealing with dental implant research these standard parameters should be assessed if possible.

We absolutely agree with this statement -as far as (long-term) survival and success studies of dental implants are concerned. Those should include the rates of biological and technical complications, clinical cases etc. The present study is none of these. It aims to present exclusively the long-term development of patients’ behaviour: Compliance with a sytematic aftercare program analogous to SPT after systematic periodontal therapy and to reveal possibly impacting factors. To the best of our knowledge, this might be the first study presenting suchlike data 1. over a 10-year observational period, 2. involving a very considerable number of patients/implants and 3. Having been conducted in a private practice setting.

Even in periodontology, comparable data concerning much longer established SPT programs have been scarce until today.

With the publication of this paper, we want to give orientation to many clinicians dealing with dental implant therapy for what they might expect as patients‘ response to their aftercare programs. Maybe for the first time they would be able to compare their long-term compliance results. If appropiate, this might leat to intensified efforts in specific communication towards the patients. Implants and patients might benefit.

To the best of our knowledge, we could hardly find studies including long-term data on intensively assessed 1. Compliance rates, 2. Biological complication rates and 3. Technical complication rates as well. Even points 2 and 3 are not always put together and point 1 has been very scarce until today.

We finally want to cordially ask this reviewer to approximate our position. We did what was possible to adress your concerns and those of the other reviewers. We hope that you now can agree with a publication.

Submission Date

17 May 2020

Date of this review

05 Jun 2020 15:02:00

Again, we want to thank all reviewers for their time and their efforts to increase the quality of our study in a considerable degree.

For all authors

Dr. Eberhard Frisch